# GrowSpace: Learning How to Shape Plants

**Yasmeen Hitti**
McGill University, Mila
yasmeen.hitti@mail.mcgill.ca

**Ionelia Buzatu**
Johannes Kepler Universität Linz
buzatuionelia@gmail.com

**Manuel Del Verme**
McGill University, Mila
manuel.delverme@gmail.com

**Mark Lefsrud**
McGill University
mark.lefsrud@mcgill.ca

**Florian Golemo**
Université de Montréal, Mila, Element AI
fgolemo@gmail.com

**Audrey Durand**
Université Laval, Mila
audrey.durand@ift.ulaval.ca

## Abstract

Plants are dynamic systems that are integral to our existence and survival. Plants are faced with environment changes and adapt over time to their surrounding conditions. We argue that plant responses to an environmental stimulus are a good example of a real-world problem that can be approached within a reinforcement learning (RL) framework. With the objective of controlling a plant by moving the light source, we propose GrowSpace, as a new RL benchmark. The back-end of the simulator is implemented using the Space Colonisation Algorithm, a plant growing model based on competition for space. Compared to video game RL environments, this simulator addresses a real-world problem and serves as a test bed to visualize plant growth and movement in a faster way than physical experiments. GrowSpace is composed of a suite of challenges that tackle several problems such as control, hierarchical learning, fairness and multi-objective learning. We provide agent baselines alongside case studies to demonstrate the difficulty of the proposed benchmark.

## 1  Introduction

Advancements in Reinforcement Learning (RL) [35] are in part from comparing algorithms on commonly used benchmarks such as the Atari Learning Environment [1]. However, doubts have been raised on popular benchmarks since they do not always translate to real-world applications and inherently fail to capture the generalization performance of RL algorithms for real-world deployment [20]. The RL community needs new simulation-driven benchmark environments with real-world properties.

Currently there are a limited number of benchmarks that represent real-world systems since they are hard to design and learning from the physical world is difficult [28, 12]. Their complexities stem from high operating costs, their slow movements, and their limited amount of data [13]. Simulators have provided a proxy to real-world systems and have demonstrated success in optimization of control tasks in robotics [33].

We direct our interest on plants, which similarly to robots, need to interact with their environment. Plants are complex and sense their surroundings through actuation and sensing systems [16]. As biological systems, they actuate their movement as a response to an external stimulus such as light [7].

Submitted to 35th Conference on Neural Information Processing Systems (NeurIPS 2021). Do not distribute.

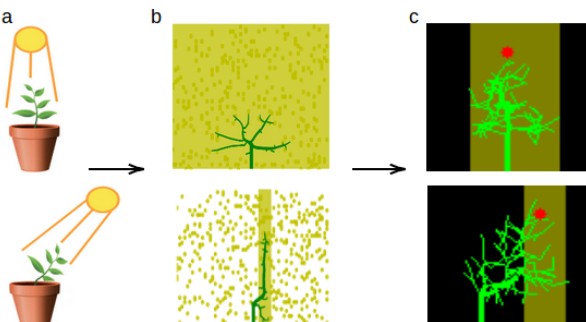

Figure 1: **High-level Overview** of the approach taken for designing the GrowSpace Environment. (a) Plants orient themselves towards light. (b) A plant branching algorithm imitates this phototropic behaviour. (c) We implemented an RL framework to reach goals (red)/shapes and enable plant growing tasks around these properties.

Their spatial reorientation and growth towards light is a tropic response because their movement is influenced by the direction of the light source [26]. Recently, the idea of controlling plant growth through light manipulation has been investigated for the development of bio-hybrid systems such as living structures [40]. The control of a biological agent, presents a set of interesting problems which translate well to the RL community, such as: continuous control [31], hierarchical learning [43], multi-objective learning [38], and fairness in a multiple plant setting [22].

In this work, we introduce GrowSpace, a new RL environment that enables the control of procedurally generated plant structures. This benchmark is based on real plant responses to light and leverages this response to address a set of diverse challenges that are beyond the scope of bio-engineering. We bring attention to a set of four different challenges that range from classic control to fairness. GrowSpace is an environment that spans across different fields such as plant science, agriculture, RL, and robotics.

The primary contributions of this paper include: (i) GrowSpace[1], an OpenAI Gym-compatible environment [5] for RL, agricultural plant science, and robotics research, (ii) the release of 4 different challenges that encompass control, hierarchical learning, fairness, and multi-objective learning, see Table 1,(iii) trained baseline agents using Proximal Policy Optimization (PPO) [34] with a CNN state encoder and a case study of the behavior and weaknesses of the agents. We do **not** claim that the environment allows for easy transfer of policies to real plants but we argue that this constitutes an important step towards more realistic RL environments, and supports developing agents for noisy biological settings.

## 2 Background

We first cover the RL framework of a Markov Decision Process (MDP), learning with fairness constraints, and learning multiple-objectives. These topics are reviewed to lay the foundation of GrowSpace and the different challenges it provides to the RL community.

### 2.1 Markov Decision Process

A MDP is a framework used to study the control of sequential decision processes for dynamic systems [30]. A MDP is represented as a tuple $\mathcal{M} = \langle \mathcal{S}, \mathcal{A}, \mathcal{R}, \mathcal{P}, \gamma \rangle$ that includes a state space $\mathcal{S}$, an action space $\mathcal{A}$, a transition function $\mathcal{P} : \mathcal{S} \times \mathcal{A} \mapsto \mathcal{S}$, a reward function $r : \mathcal{S} \times \mathcal{A} \mapsto \mathbb{R}$, and a scalar discount factor $\gamma$. For each time step $t$, a RL agent is in a state $s_t \in \mathcal{S}$, interacts with the environment and chooses an action $a_t \in \mathcal{A}$ which leads to a reward $r_t \sim r(s_t, a_t)$ and transitions to a new state $s_{t+1} \sim \mathcal{P}(s_t, a_t)$. The goal of a RL agent is to learn a policy $\pi : \mathcal{S} \times \mathcal{A} \mapsto [0, 1]$ such as to maximize the discounted sum of rewards.

---

[1] `https://github.com/YasmeenVH/growspace`

## 2.2 Fairness in RL

Fairness is of concern in RL when actions selected by the agent affect the state and latter rewards. In a MDP setting, several constraints of fairness have been introduced over the past years. In the multi-armed bandit learning framework, fairness has been studied in the setting where the selection of an arm with lower expected reward over another arm is considered unfair [23]. Jabbari et al. [22] implement this constraint in an MDP setting, stipulating that in any state $s$, an algorithm cannot favor action $a$ that has a lower probability of a expected reward than action $a'$. Wen, Bastani, and Topcu [41] propose fairness constraints that provide equality of opportunity [19] and have observed that parity between groups reduces rewards more than equal treatment.

## 2.3 Multi-objective RL

Multi-objective reinforcement learning involves having two or more objectives that may conflict with each other and need to be achieved by an agent [37]. Rewards in this context are a feedback signal that are represented as a vector of length equivalent to the number of objectives to attain [6]. Conflicts amongst objectives are observed when certain objectives are being favored over others. To reduce conflicts, trade-offs are used between objectives. The most widely used optimality criterion is the Pareto dominance relation [38]. Pareto dominance happens at the policy level, when a policy surpasses all other policies for all objectives. Learning policies that meet all preferences has been shown to be a challenging task and consequently the problem is often reformulated as a single-objective problem in the literature [42]. This comes with limitations because certain behaviours can emerge and show preferences to one of the objectives.

# 3 Related Work

The proposed GrowSpace environment complements current RL benchmarks and existing plant modelling platforms.

## 3.1 RL Benchmarks

The Arcade Learning Environment (ALE) [1] has long been used as a benchmark for evaluating AI agents on a variety of tasks. These tasks have pushed our knowledge and the direction of research notably in representation learning, exploration, transfer learning, model learning, and off-policy learning [27]. Similarly, StarCraft II [39] presents harder tasks than prior video game-based environment. However, as mentioned earlier the usage of common benchmarks has been put into question and how they could translate to the real world [20]. Recently, interest has been pushed on procedurally generated environments such as Procgen Benchmark [10] and the NetHack Learning Environment [24] both with the intent of tackling generalization with large amount of tasks and levels. The focus of these benchmarks are not real world-orientated. The closest RL benchmark to real-world interaction is Mujoco [36], a physics engine that enables testing of robotic simulations with contacts. Although Mujoco can adapt different types of bodies and movements, no task formulation has been addressing a greater challenges such as fairness. GrowSpace fills this gap.

## 3.2 Plant Modeling

Plants are interesting subjects to simulate as they are self-organizing systems that have the ability to adapt to dynamic environments by sensing their surroundings and directing their growth to preferable regions [8]. Plant models have evolved throughout the past two decades and have been incorporating the effects of environmental conditions [4]. Simulation of realistic virtual plants and trees have been explored through different algorithms such as L-systems [29], Functional–Structural Models (FSMs) [11] and Space Colonization Algorithms (SPA) [32]. Plant modeling has received increased interest and has primarily focused on: the reconstruction of plant architectures overtime, discovery of underlying ecophysiological mechanisms driving certain plant traits, and the movement of nutrients and their allocation throughout the plant body [14]. The development of these models are beneficial to understand the functioning, manipulation and hypotheses of plant growth. However, they are not feasible for generating and controlling behavioral patterns that a plant may exhibit [3]. We're basing our simulator on the Space Colonization Algorithm, adding a controllable light source and target points and shapes for the plants to grow towards.

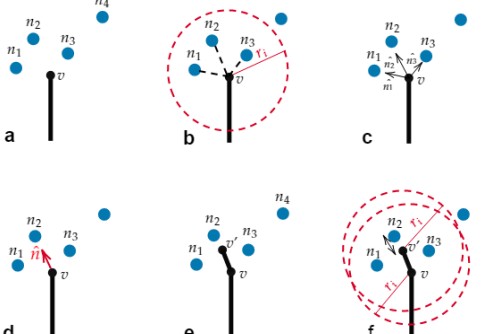

Figure 2: Steps for branching in the Space Colonization Algorithm, where (a) all photons are filtered (b) trough a radius of attraction (c) and their normalized vectors from the plant tip to the photons (d) are summed and normalized to find the direction of growth (e) for the new plant segment to be attached (f) process is repeated for all existing plant tips

## 4   GrowSpace Learning Environment

We present GrowSpace, a new procedurally generated RL environment that is built with the OpenAI gym interface [5]. The simulator is inspired by a real-world problem of optimizing plant physiology and direction of growth over time. In the real-world, plant growth is dictated by several variables, an important one is light availability. GrowSpace incorporates a plant's behavioral response to light and provides control over the branching by means of a mobile light (either beam light or small point light). The objective is to guide the growing plant to a desired target or shape depending on the challenge. Figure 1 provides an overview of our approach for designing GrowSpace. Much like in the real world, the light directly influences the direction of growth of a plant (1a). A branching algorithm is chosen to mimic a plant's relationship to light (1b). Finally, the branching algorithm is formulated as a RL problem where an agent's objective is to shape a plant towards a target (red) or a desired configuration through means of a mobile light (1c).

### 4.1   Plant Branching

The Space Colonization Algorithm (SCA) [32] is implemented for simulating the branching at each time step in GrowSpace. Through the attachment of plant segments to a plant structure, this algorithm facilitates the iterative growth of a virtual plant. The direction of growth is determined by the location of the attraction points. In GrowSpace, to imitate phototropic behaviour of a plant, the attraction points are thought of as available photon particles. To avoid predetermined shapes in GrowSpace, the photon particles are scattered at random to facilitate stochastic branching. The number of particles are user-defined and can be compared in a real life setting to the available light intensity: the higher the light intensity, the greater the density of photon particles, the more branching occurs.

Figure 2 (inspired by [32]) illustrates the algorithm. The algorithm begins with a set of photon particles $N$ and an initial plant segment with tip $v$ (a). The plant segment tip eventually become a set as the plant grows, where $v \in V$. In order for a plant tip to grow, photons $n \in N$ must be located within a predefined radius of influence $r_i$, as seen in (b) where $n1$, $n2$ and $n3$ attract segment tip $v$. When a photon is too close to a plant segment, the photon is removed and is not considered. The normalized vectors from tip $v$ towards photons $n \in N$ are computed (c). Once summed, the normalized vector $\hat{n}$ is found for $v$ (d). The vectors representing the direction of growth are:

$$\overrightarrow{n} = \sum_n^N \frac{n-v}{\|n-v\|}.$$

(1)

The final normalized vector for a plant segment tip is: tip is:

$$\hat{n} = \frac{\overrightarrow{n}}{\|\overrightarrow{n}\|}.$$

(2)

Vector $\hat{n}$ represents the direction of growth of plant segment tip $v$ which is towards photon $n_2$ in this example. The plant grows a new segment $v'$ of a length that is user determined and fixed throughout the plant growth (e). The procedure is then repeated on both of the plant segment tips (f), we can observe that $n_2$ is too close to $v'$ and will not be considered for branching.

Simulations of the space colonization can vary due to initial configurations chosen by the user (see Appendix A. In GrowSpace, we limit the amount of observable photons to the plant with a light source. The light source illuminates photons within a certain range, this consequently restricts the direction of growth. We introduce the concept of light direction in order for the artificial plant to grow unidirectional towards the light source. To grow towards the light source, shading needs to take place as to not allow the light beam to illuminate the photons that are below existing parts of the plant foliage. These hypotheses are based on phototropism, a response process, that enables plants to adjust their growth towards the direction of the light [17].

## 4.2 Reinforcement Learning Framework

We formulate GrowSpace as a MDP described by a state space $\mathcal{S}$ that is accessed by the agent as a pixel observation, an action space $\mathcal{A}$ that can be discrete or continuous, a transition function $\mathcal{P}$ and a reward function $R$. On each time step $t$ of a learning episode, the agent observes the state $s_t \in \mathcal{S}$, takes an action $a_t \in \mathcal{A}$, moves to a new state $s_{t+1} \sim \mathcal{P}(s_t, a_t)$, and receives a reward $r_{t+1} \sim R(s_t, a_t, s_{t+1})$. The probability of a plant segments tips to branch in a specific direction given action $a_t$ in state $s_t$ is incorporated into the transition probability $P(s_{t+1}|s_t, a_t)$. In this environment, much like in the real world, the light directly influences the direction of growth of a plant. The agent's objective is to shape a plant towards a target or a desired configuration through means of a mobile and adjustable light source.

**States and Observations:** For every step taken in the environment, the agent observes the *observation* of its current *state* prior to selecting an action. Once an action is selected by the agent, the new state becomes the observation for the next time step. States and observations are an image representation of the environment which display the plant structure, the light source and the target. The observations are available to the agent as an RGB image that contains the plant, the target and the light beam at time step $t$. The dimensions are of of $84 \times 84 \times 3$ , except for the plant shaping challenge where the dimensions are of $28 \times 28 \times 3$.

**Actions:** GrowSpace provides a discrete action space and a continuous action space. In the discrete action space the agent can execute five discrete actions. The agent can move the light beam to the right, the left or stay put. The agent can equally increase or decrease the available light beam to the plant. The movement of the light beam is set at a default of 5 pixels in any given direction and can be customized by the user. The continuous action space has two actions, the light velocity, speed at which the light is displaced, and the width of the light beam. This could be a more realistic and more complex set-up, and it will help to transfer the problem from simulation to real world. The actions chosen will influence the available scattering to the plant and will impact the direction of growth of the plant. For example, if the beam of light is not close enough the plant will not be able to branch out because the attraction points and will be dormant. If the light reveals several points, branching will be occur in multiple places in the illuminated area.

In the multiple-objective task, the action set changes due to the circular light beam. Similarly, the agent can increase or decrease the light beam radius, it can move left and right and, can move up and down. The default radius of the beam is 10% of the width of the environment.

**Rewards:** The reward will be dense and will be received at each time step. Rewards will depend on the challenges in which the agent is trying to solve. Rewards are task specific and explained below in Section 5.

**Episode and Reset:** The episode length is fixed and is set to 50 steps. At the beginning of each episode, the scattering of photons, and the initial plant stem, as in Section 4.1, and the target(s) are procedurally generated in order to ensure the agent will not have visited the exact state previously in other episodes.

## 5 Tasks

We propose an initial set of tasks that can be tackled in the GrowSpace environment, all of which with several levels of difficulty. The combination of tasks released encompass some known challenges to the RL community, such as control, hierarchical learning, fairness, and multi-objective learning. Table 1 provides an overview of the tasks and their respective challenges.

| | Challenges | | | |
|---|---|---|---|---|
| Tasks | Control | Hierarchy | Fairness | Multi-objective |
| Grow Plant to Goal | ✓ | ✓ | ✓ | ✓ |
| Find Plants | | ✓ | ✓ | ✓ |
| Grow Multiple Plants | | | ✓ | |
| Grow Plant to -Shape | | | | ✓ |

Table 1: Reinforcement learning challenges arising from each task within GrowSpace.

**Grow Plant to Goal:** The task consists in growing the plant with the light beam towards a target positioned at random in the upper 25% of the environment. Every episode begins with the light beam positioned above the original plant stem. The agent must displace the light beam to control and direct the growth of the plant towards the target. After each action, the agent is rewarded based on the smallest distance between any of the branch tips and the target. Let $d_{b,g}$ denote the Euclidean distance between a branch tip $b$ and a target goal $g$:

$$d_{b,g} = \sqrt{(x_b - x_g)^2 + (y_b - y_g)^2}. \tag{3}$$

The reward obtained at time step $t$ is inversely proportional to this distance of the branch tip closest to the goal among the current branch tips $\mathcal{B}_t$:

$$R_t = \max_{b \in \mathcal{B}_t} \frac{1}{d_{b,g}}. \tag{4}$$

Rewards are therefore in the range $]0, 1[$. This typical control problem [31] is considered the simplest of the tasks since the light movements directly impact the plant from the beginning of the episode. The difficulty of this task is proportional to the distance between the target and the original plant stem tip; as the distance increases, the harder the task becomes.

**Find Plant:** The task consists in finding the original plant stem with the light source, either the beam or circular light. An episode starts with the light source and the original plant stem positioned at different random locations in the environment. This becomes a hierarchical learning problem [43] where the agent has to first locate the original plant stem by displacing the light source in order to increase the reward signal. The reward is computed using Equation 4. The difficulty of this task is proportional to the distance between the target and the original plant stem tip (as in the Grow Plant task), and to the distance between the original plant stem and the initial light source position. Displacing the light source multiple times before finding the plant reduces an agent's ability to attain the highest amount of rewards.

**Grow Multiple Plants:** The task consists in finding two or more plant stems with the light beam and growing them to similar maturities throughout the episode. In this task, the agent must grow $n > 1$ plants towards a target. The target is placed at random in the upper 25% of the environment, the light beam and initial plant stems are initialized randomly within the environment. As in the Find Plant task, the agent must displace the light beam to find all the existing plants in order to initiate a reward signal. The reward consists in the minimum distance reward (Eq. 4) over all plants:

$$R_t = \min_{1 \le i \le n} R_t^{(i)}, \tag{5}$$

where $R_t^{(i)}$ is the grow plant reward (Eq. 4) associated with plant $1 \le i \le n$. As seen in Section 2.2, different fairness constraints can be adopted in a MDP setting and could be integrated within GrowSpace. We set our first fairness task with a fairness constraint that is similar to [41], which suggests that the agent should provide equal opportunity for each plant to grow towards the target at every step of the episode. The difficulty of this task is in sharing the amount of available photons adequately between plants when they start growing closely to each other. As different plants start approaching each other the photons may run out in the desired direction of the target and the plants may never reach the target (see Appendix F).

**Grow Plant to Shape:** This task consists in growing plants into specific shapes by using a circular light source that can navigate to precise locations in the environment. As default shapes for benchmarking purposes we consider the MNIST dataset [25], which is widely used in machine learning.

MNIST contains $28 \times 28$ pixel binary images of handwritten digits (0-9). Given an MNIST image, the goal is to grow a plant such that its shape matches the drawn digit as best as possible. For this task, the environment is reshaped to a width and height of $28 \times 28$ pixels (i.e. the size of a MNIST image). The agent has to grow the plant into multiple directions to best cover the outline of the MNIST digit without growing out of bounds. This is a multi-objective task, since the agent has to cover multiple areas in any order, while also keeping the overall goal of limiting the amount of branching in mind.

The reward for this task is crafted using the Jaccard Index [15] similarity score. Let $\mathcal{A}_t$ and $\mathcal{G}$ respectively denote the set of pixels that the plant occupies at time steps $t$ and the set of pixels that belong to the target shape. The reward at time step $t$ is given by the similarity score:

$$R_t = \frac{\mathcal{A}_t \cap \mathcal{G}}{\mathcal{A}_t \cup \mathcal{G}}. \tag{6}$$

## 6 Experiments and Results

We demonstrate in this section how GrowSpace presents a set of challenging tasks for RL algorithms through a set of case studies.

**Baselines:** We evaluated several gradient-based policy methods in the general control setting (Grow Plant task): Proximal Policy Optimization (PPO) [34], Advantage Actor Critic (A2C) [18], and Rainbow DQN [21]. The plots of average reward per episode can be found in Appendix C.

For each of these agents, a state is represented by a tensor of $(3, w, h)$ where $w$ and $h$ are the width and height of the observed image in the task. These representations are fed through three convolutional layers, a fully connected layer and a final layer using the ReLU activation function. The output of the policy network is a probability of each action belonging to the action space. Results obtained on the Grow Plant task indicated that PPO was the most promising strategy for this problem (see Appendix C). We therefore conducted a hyperparameter search for PPO across all challenges and with three different seeds. The details of the final chosen PPO parameters can be found in Appendix B.

A random agent and an oracle agent have also been implemented. No training was performed for the random and oracle agents. The random agent selects actions uniformly at random from the action space. For each challenge, a unique oracle agent is implemented. More information about the oracle solutions can be found in Appendix D.

**Performance metrics:** To understand if learning is successful, we compare the mean episodic reward as our performance metric. To better interpret the agent's behaviour, we include other metrics such as the selection of actions and the overall number of branches produced throughout an episode. Results are always averaged over three runs (different random seeds).

### 6.1 Case Studies

We present a set of case studies to display a spectrum of behaviors the agent can display and where challenges are shown to be difficult. Each case study consists in one easy and one hard configuration (in terms of difficulty), to be described below. Figure 3 shows the cumulative rewards (averaged and one standard deviation) for the three baselines in easy and hard configurations of each case study.

**Control:** We define an easy setting when the target is above the original plant stem (Grow Plant task) and a hard setting when the stem and target are at opposite extremities of the environment (Find Plant task).

Figure 3(a) shows that the easy control reward curve from PPO is closer to the oracle solution and that learning can be improved. The hard control challenge is indeed more difficult as the highest reward achieved by the oracle is much lower than in the easy setting. For both difficulties we observe that the PPO reward curve is midpoint between the oracle and random action selection, suggesting that PPO's behaviour can be optimized further. The video renderings show the agent displacing the light away from the plant to quickly, loosing steps with stagnating rewards instead of growing new closer branches and does not succeed in guiding the plant to target. Equally, the episodic action selection as seen in Figure 6(a) in Appendix D demonstrates that agent does not favor decreasing the light beam resulting in a plant with multiple branches competing for the same photons in the direction of the target and thus resulting in slower growth and lower rewards. The action distribution in the easy case

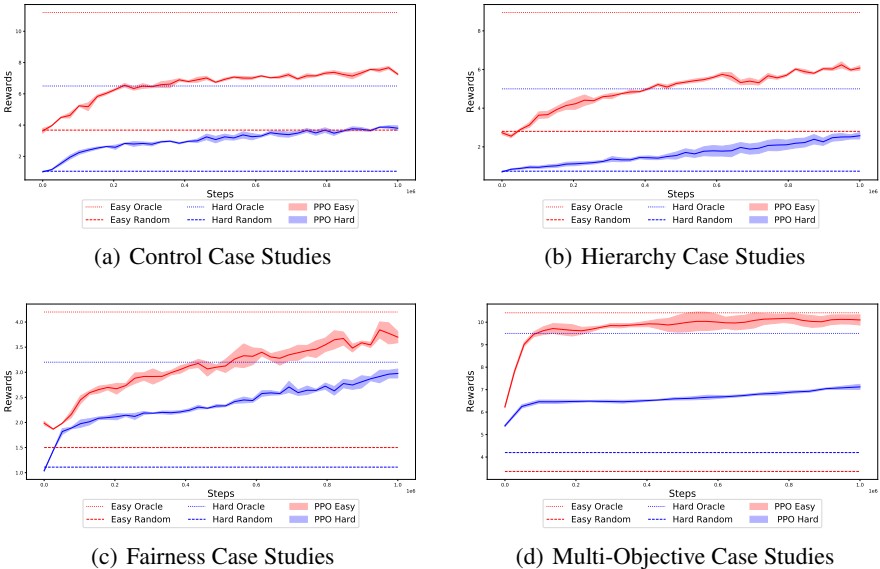

(a) Control Case Studies  (b) Hierarchy Case Studies

(c) Fairness Case Studies  (d) Multi-Objective Case Studies

Figure 3: **PPO Baseline Performance.** For each environment variation, we are plotting the lower bound (random baseline) and upper bound (oracle), as well as the performance of a PPO agent.

is relatively similar amongst actions, however in the hard setting it is noticeable that the right and left actions are used more (see Figure 6 in Appendix D). This can be explained as the plant does not need to simply grow vertically but laterally to the opposite side of the environment.

**Hierarchical Learning:** We present two case studies similar to control with the exception that the initialisation of the episode starts with the light placed at random and not above the plant as the agent needs to first find the plant.

Figure 3(b) shows that the hard hierarchy reward curve from PPO yields a smaller amount of rewards. Similar to control, the hard setting has a lower reward due to the distance between the initial plant stem and the target. With the initial task of finding the plant first, the low reward in the hard setting can be explained by the agent receiving the same reward while trying to find the plant and, the greater distance between the target and the initial stem. The action of increasing the light is more utilized within the harder setting to find the initial plant stem (see Figure 7 in Appendix D). With the video renderings, we also see that the light width is not changed dramatically once the initial stem is found and the agent learns to drag the light towards the target. The video renderings equally show that the plant gets bushy and the smaller light width is not utilized efficiently to try and reduce competition amongst branches for available photons (see Appendix F).

**Fairness:** We present two case studies. For the hard setting, the initialization of an episode starts with the plants at the opposite extremities of the environment and the target is placed in the middle of the environment. For the easy case study, the episode initialization starts with both plants at a distance that is set to the default light width and the target is in the middle. This case study is particular because the plants are very close and competing for available photons in order to reach the target. As a fairness challenge, the objective is to produce plants of similar size.

Figure 3(c) shows that the easy fairness reward curve from PPO produces the highest amount of rewards. Both PPO reward curves are between closer to the oracle bound than the random agents for both cases. We investigate if the agent's behaviour is fair by looking at the median amount of branches per plant, where the numbers are relatively close (see Figure 8 in Appendix D. The easy case produces a smaller amount of branches, this can be explained by the small pool of photons that are available to both plants branching and thus limiting additional branching in the right direction. In the middle case, the branching is higher and can be explained by the greater amount of photons available to both plants while reaching the target as they do not need to compete for the majority of the episode.

**Multi-objective Learning:** We first compare all Mnist digits to better understand the proposed challenge. The digits are compared by their median reward values from PPO as seen in 4(a). The order of the digits presented in the curriculum from easiest to hardest is 3, 6, 2, 1, 4, 5, 7, 8, 9, 0. The curriculum consists of 2000 episodes with the two first easiest digits and for every increment of 2000 episodes a new new digit is added. The last 6000 episodes of training have all the MNIST digits.

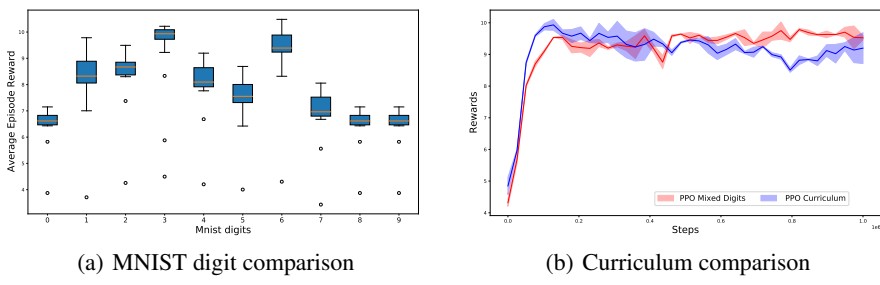

(a) MNIST digit comparison  (b) Curriculum comparison

Figure 4: Comparison of digits to design the curriculum for training

In Figure 4(b) the learning seems at a higher rate in the first episodes of training for the curriculum approach however, the reward curve decreases as the addition digits are added. The random selection of digits seems to be a better fit over time. We can see that the agent is focused on density on plant vs overall shape as the light width fluctuates a lot in the video renderings but it does not visit the full trajectories of the MNIST digits.

## 7   Limitations

The limitations of GrowSpace are translating plant growth control into practice. The benchmark provides a modest first step to modeling a plant response that occurs in the physical world however, under the assumption of all other environmental conditions being constant (water supply, wind, nutrient availability, etc). The transfer of an optimal policy in simulation may not succeed when reproducing the experiment in the real-world however, high-level intuition can be extracted from the optimal policies [9]. GrowSpace implements one plant model for a generalized perspective into plant growth, specific models for different plant species could enable better precision and simulations that are specific to researchers needs.

## 8   Conclusion and Future Work

GrowSpace is a procedurally generated environment with a set of challenges that can help the advancement of reasearch in RL and agriculture. It encompasses real-world behaviour of plants in a low representation setting and provides a series of challenges that address issues such as fairness. We provide gradient based agent baselines for the control challenge to display the difficulty of the easiest challenge within GrowSpace. Case studies with our base performing baseline, PPO, are layed out to give insights on the type of behaviour an agent can adopt in easy and hard settings. We demonstrate that indeed GrowSpace is a environment that is complex and proposes different settings which enable different skills to be learnt such as sharing ressources in the fairness constraint, patience for displacing a light to grow the plant and limiting available resources to a growing plant for precision.

Further add-ons can be attainable in order to recreate a full growing environment dynamic with water, nutrients, wind and even specific plant models. We plant to support GrowSpace after its release as well as introduce new environment parameters. In sum, plant growth is a grounded and intricate topic and its full control is not fully understood. GrowSpace provides a first step in the direction of plant growth control through a known plant response, phototropism.

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
