# OpenReview forum: "GrowSpace: Learning How to Shape Plants"
_NeurIPS.cc/2021/Track/Datasets_and_Benchmarks/Round1 — Submitted to NeurIPS 2021 Datasets and Benchmarks Track (Round 1)_

### Official Review · Reviewer_MbQi · 2021-07-03
**An interesting problem setting but perhaps too narrow to be generally useful**

**Rating:** 5
**Confidence:** 5

**Strengths:**

- An interesting problem domain treating plants as dynamical systems and the tasks of shaping them properly to grow habitable spaces
- Well written paper, clear to understand and thorough in their claims and experiments
- Authors are modest about the claims, highlight the weakness, and are candid about the shortcomings. I really cherished your points. #kudos

**Weaknesses:**

- Unfortunately the problem space too narrow to be generally useful to the community
- If I instead evaluate the main contribution to be around benchmarks to evaluate hierarchal learning, fairness, and multi-objective learning, the scope and the technical rigor aren't up to the mark to carefully evaluate the fundamental issues of these domains
- The original motivation of providing real world tasks is well justified, but as the authors themselves noted -- neither the simulations nor the final policies are validated by real world data or experiments. Additionally the underlying problem studied is an over simplification of the real world problem.

**Additional Feedback:**

- I really like how candid authors are with their claims, limitations, and weaknesses. I hope to see more researchers will set similar examples to follow upon.
- One of the most interesting problem setting I have seen in RL benchmarking lately.

**Clarity:**

Over all the paper is well motivated and presented. A few suggestions below
- It was hard to initially follow how different tasks cater of different challenges. Possibly providing some context while introducing the task could help.
- Line 317 - How is this order established?
- Unclear what the two kinds of observations are and why they are useful, perhaps a picture can say a thousand words here


**Correctness:**

- I found the experiment designs and evaluations to be well motivated and designed
- Line 95-96: The claim that  robotics simulations with MuJoCo can't encompass challenges with fairness and multi-objectivity is just not true. On one hand these challenges aren't modelling issues but task formulation issues. On the other hand, there are multiple well established tasks in the RL community that deals with multi-objective problems (almost all RL continues control tasks are multi-objective [1][2])

[1] Learning Complex Dexterous Manipulation with Deep Reinforcement Learning and Demonstrations, Rajeswaran, Kumar et al.
[2] https://meta-world.github.io/





**Documentation:**

- yes

**Ethics:**

- No. All results are in simulations

**Relation To Prior Work:**

- paper can use a bit better positioning with respect to continuous control reinforcement learning literature and benchmarking.

**Summary And Contributions:**

Paper presents an interesting problem domain treating plants as dynamical systems, formulates benchmark tasks around them to study control, hierarchical learning, fairness and multi-objective learning. Authors also provide baselines and case studies.

---

> ### Author Response · Authors · 2021-07-13
> **Response to reviewer 3**
>
> Thank you R3 for the time and effort that went into your review.
>
> **Regarding “problem space too narrow to be generally useful to the community”:**
> Simple benchmarks have been widely used amongst the RL community, like `gym-minigrid`, i.e. BabyAI’s gridworld [1], which has 34 citations and is an established environment for developing new discrete action RL agents. Similarly, we have developed a benchmark that is fast to use and easy to understand, but poses significant problems to contemporary DRL methods.
>
> **Regarding “scope and the technical rigor aren't up to the mark to evaluate the fundamental issues of fairness, hierarchical learning and multi-objective learning”:**
> We are very surprised by this comment, as we do provide a wide range of rigorous experiments showing clear cases of failure for the baselines. Could you be more explicit about what kind of experiments / rigor are missing? Our work does not attempt to present new agents that can solve the aforementioned problems, but rather point out that this simple setting is enough to pose significant hurdles for SOTA DRL methods. We are merely creating a fertile ground for future development of these algorithms.
>
> **Regarding comments about clarity:**
> - **”It was hard to initially follow how different tasks cater of different challenges. Possibly providing some context while introducing the task could help.”** Table 1 summarizes the challenges found within each task. But thank you for the remark, we have added the sentence at line 231: “ The agent has to grow the plant into multiple directions to best cover the outline of the MNIST digit without growing out of bounds. This is a multi-objective task, since the agent has to cover multiple areas in any order, while also keeping the overall goal of limiting the amount of branching in mind.”
> - **”Line 317 - How is this order established?”** The order of the MNIST digits is established by comparing the mean reward of every digit. The digits are in order from highest reward to lowest amount (easy to hard). Please refer to figure 4(a) where mean rewards are found for every digit. The text has been changed to: “The digits are compared by their median reward values from PPO as seen in Figure 4(a).”
> - **”Unclear what the two kinds of observations are and why they are useful, perhaps a picture can say a thousand words here”** Thanks for pointing this out. Since all testing has been done with one type of observation, we have decided to remove the second type (binary). Section 4.2 was updated accordingly.
> - **”Line 95-96: The claim that robotics simulations with MuJoCo can't encompass challenges with fairness and multi-objectivity is just not true.”** We thank you for your comment and have changed the sentence in text to: “Although Mujoco can adapt different types of bodies and movements, no task formulation has been addressing greater challenges such as fairness.”
>
> **References:**
>
> [1] Chevalier-Boisvert, Maxime, Willems, Lucas, and Pal, Suman. “Minimalistic Gridworld Environment for OpenAI Gym”. Github, 2018. https://github.com/maximecb/gym-minigrid

---

### Official Review · Reviewer_Qzkg · 2021-07-03
**A novel environment that connects agriculture with AI**

**Rating:** 7
**Confidence:** 3
**Clarity:** Yes the paper is well written

**Strengths:**

The paper is well-written, and the idea of creating a benchmark for evaluating the growth of the planet is novel and beneficial to the agriculture and AI community.

The environment is open-sourced and based on OpenAI gym, so the users can easily experiment with many baseline RL algorithms.

The environment leverages the Space Colonization Algorithm for the branching based on the light source, which can simulate the branching in the (ideal) real-world environment.

The environment is evaluated with baseline RL algorithms and also a case study.

**Weaknesses:**

The simulation is still very simple, and doesn't support multiple variables in real life such as temperature, light strength, wind etc.

As a simulation, it would be more persuadable if the author can provide real-world experiments, even with the simplest settings.

**Additional Feedback:**

na

**Correctness:**

Yes, the simulation is based on Space Colonization Algorithm for branching, and the metrics/rewards for the 4 tasks Grow Plant to Goal, Find Plants, Grow Multiple Plants and Grow Plant to Shape all look correct to me.

**Documentation:**

yes, the proposed environment is open-sourced at https://github.com/YasmeenVH/growspace

**Relation To Prior Work:**

Yes

**Summary And Contributions:**

The paper proposed GrowSpace, a RL environment based on OpenAI gym that enables the control of procedurally generated plant structures. The provided benchmark can simulate the branch based on photons provided by the light source, so users can leverage this environment to train smart agents for growing planets towards specific goals.

The proposed benchmark has 4 different challenges including encompass control, hierarchical learning, fairness, and multi-objective learning. And the authors provide baseline agents using Proximal Policy Optimization (PPO) and a case study of the behavior and weaknesses of the agents.

---

> ### Author Response · Authors · 2021-07-13
> **Response to reviewer 2**
>
> Thank you R2 for the thoughtful review.
>
> **Regarding “Simulation is simple and does not consider other environmental variables”:**
> We indeed aimed for a simpler simulation, making sure that the system would be realistic enough to encourage cross pollination between Reinforcement Learning and Bioengineering, yet computationally efficient to avoid creating a barrier of entry for less-funded researchers. We did consider more realistic models including weather and water [1], but the added realism did not justify the additional computational burden.
>
> **References:**
>
> [1] We implemented a version of the environment that features an additional controllable water level, which is available here (in a branch of the original repo) https://github.com/YasmeenVH/growspace/blob/RockyPlantsExperiments/growspace/envs/growspace_resources.py

---

### Official Review · Reviewer_w4Wf · 2021-07-05

**Rating:** 5
**Confidence:** 3
**Correctness:** Yes
**Clarity:** Yes

**Strengths:**

The paper presents an OpenAI Gym-compatible environment for using RL to grow plants, which makes it easy for the community to use. The proposed benchmark tackles a real-world problem, which is of great significance to real-world applications of RL algorithms.

**Weaknesses:**

The proposed benchmark is a bit artificial, especially the tasks on growing plants into shapes of MNIST digits. All of the environments are constructed in simulation, which makes me uncertain about the impact on growing plants in the real world. Moreover, the evaluations only consider PPO, which seems rather limited.

**Additional Feedback:**

See above

**Documentation:**

Yes

**Relation To Prior Work:**

Yes

**Summary And Contributions:**

This paper proposes a new method for RL in learning to shape plants. The benchmark proposes several tasks in growing plants that involve challenges in control, hierarchical learning, fairness, and multi-objective learning. The authors evaluate PPO on the proposed environment in the four problems described above.

---

> ### Author Response · Authors · 2021-07-13
> **Response to reviewer 1**
>
> We would like to thank R1 for their time and effort in composing this review.
>
> **Regarding “the evaluations only consider PPO”:** In the original submission, we included evaluations on 3 stable DRL baselines: Rainbow, A2C, and PPO. The results of these experiments are included in the appendix C. In these experiments, we found that PPO surpassed the other 2 consistently, which is why we moved them to the appendix and only included PPO in the main body of the paper. Please refer to line 247 in the main paper.
>
> **Regarding the “benchmark is a bit artificial”:** While there are many lines of work pushing the realism in simulated environments, especially for robot navigation, in this work we focus on developing a simple, fast, and easy-to-use benchmark that is useful in developing novel RL algorithms. Not despite but because of its simplicity, the famous “gym-minigrid” environment [1] has enabled the rapid development of new DRL agents, and we hope that similarly, by being performant but hard-to-master, our environment will have a comparable impact.
> We would also like to point out that arbitrary shapes such as ones from MNIST have been used for similar drawing tasks in the ML community. Gregor et al. [2], and Huang et al. [3] are good examples of utilizing the MNIST dataset as a benchmark dataset on tasks that then could be translated into real-world such as drawing and painting in this case.
> MNIST digits are a familiar and valid first milestone into exploiting plant phototropic responses. Our results already show that this problem highlights some issues in SOTA RL methods.
>
> **References:**
> [1] Chevalier-Boisvert, Maxime, Willems, Lucas, and Pal, Suman. “Minimalistic Gridworld Environment for OpenAI Gym”. GitHub, 2018. https://github.com/maximecb/gym-minigrid
>
> [2] Gregor, Karol, et al. "Draw: A recurrent neural network for image generation." International Conference on Machine Learning. PMLR, 2015.
>
> [3] Huang, Zhewei, Wen Heng, and Shuchang Zhou. "Learning to paint with model-based deep reinforcement learning." Proceedings of the IEEE/CVF International Conference on Computer Vision. 2019.

---

### Author Response · Authors · 2021-07-13
**Common response with respect to GrowSpace's impact on real-world and simplicity.**

We would like to thank the reviewers for their concerns and comments about the potential impact of this environment in the real-world. Since this topic was raised by all reviewers, we wanted to make it a shared discussion and clarify some points.

First, the plant branching algorithm used in our simulator (i.e. the Space Colonization Algorithm) has been used in the plant science community and the gaming community for the rendering of realistic trees [1][2]. Despite its simplicity, it is an established model of plant development.
Our implementation of the Space Colonization Algorithm further integrates real-world components such as plant phototropic responses. In our settings,  the agent can manipulate the environment through light displacement and exploit the phototropic behaviour of a plant. The control of plant growth is of interest to several fields such as bio-hybrid robotics and plant science [3] [4]. More generally, exploiting biological behaviour can lead to the design of living structures and more precise forms of agriculture in controlled environments. The GrowSpace simulator is a first step in investigating the potential use of plants for such applications by providing a fast testbed.

Although we are aware of the simplicity of the resulting simulator, a plant’s precise simulation with multiple environmental variables is not feasible yet as it is time-consuming and not all environmental interactions have been isolated. A general branching grammar provides an adequate level of abstraction and a means to validate if plant control through light displacement is possible over a longer period of time. Von Mammen et al. [5], argue that for bio-hybrid systems, when biology and engineering are mixed, evaluating the growing pattern of an individual plant is sufficient in order to understand how it influences the overall system. Finally, even if plant phototropic responses means it will always grow towards the light, strategies may differ by plant species.

**References:**

[1] Ratul, Rafsan, et al. "Applicability of space colonization algorithm for real time tree generation." 2019 22nd International Conference on Computer and Information Technology (ICCIT). IEEE, 2019.

[2] Chaudhury, Ayan, and Christophe Godin. "Skeletonization of plant point cloud data using stochastic optimization framework." Frontiers in Plant Science 11 (2020): 773.

[3] Heinrich, Mary Katherine, et al. "Constructing living buildings: a review of relevant technologies for a novel application of biohybrid robotics." Journal of the Royal Society Interface16.156 (2019): 20190238.

[4] Wahby, Mostafa, et al. "Autonomously shaping natural climbing plants: a bio-hybrid approach." Royal Society open science 5.10 (2018): 180296.

[5] von Mammen, Sebastian Albrecht, et al. "Interactive simulations of biohybrid systems." Frontiers in Robotics and AI 4 (2017): 50.

---

### Author Response · Authors · 2021-07-14
**Available for further questions**

We would like to thank our reviewers again and let them know we are aware that we provided a late rebuttal in the discussion phase.  In case the reviewers have any last-minute questions or remarks that we didn't address in the rebuttal, we will be available until the very end of the rebuttal period to respond.

---

### Decision · Program_Chairs · 2021-07-26

**Decision:**

Reject

**Comment:**

While this paper proposes an interesting setting of growing a shape plant, reviewers agree that it is unclear how this simplistic setting with no real world data would add to the community. While simplistic settings are used by many, further distinction from ones already provided by OpenAI gym would be necessary. Reviewers also argue that in order to overcome this, even a simplest setting real-world data or experiments could bring this work a lot further. We also recommend mentioning other evaluations authors did (but moved to appendix) into the main text to avoid confusion.